# *RALGAPA1* Deletion in Belgian Shepherd Dogs with Cerebellar Ataxia

**DOI:** 10.3390/genes14081520

**Published:** 2023-07-25

**Authors:** Matthias Christen, Isabel Zdora, Michael Leschnik, Vidhya Jagannathan, Christina Puff, Enrice Hünerfauth, Holger A. Volk, Wolfgang Baumgärtner, Tessa C. Koch, Wencke Schäfer, Miriam Kleiter, Tosso Leeb

**Affiliations:** 1Institute of Genetics, Vetsuisse Faculty, University of Bern, 3001 Bern, Switzerland; matthias.christen@unibe.ch (M.C.); vidhya.jagannathan@unibe.ch (V.J.); 2Department of Pathology, University of Veterinary Medicine Hannover, 30559 Hannover, Germany; isabel.alexandra.zdora@tiho-hannover.de (I.Z.); christina.puff@tiho-hannover.de (C.P.); wolfgang.baumgaertner@tiho-hannover.de (W.B.); 3Center of Systems Neuroscience, Hannover Graduate School for Neurosciences, Infection Medicine, and Veterinary Sciences (HGNI), 30559 Hannover, Germany; 4Department for Companion Animals and Horses, University of Veterinary Medicine Vienna, 1210 Vienna, Austria; michael.leschnik@vetmeduni.ac.at (M.L.); miriam.kleiter@vetmeduni.ac.at (M.K.); 5Department of Small Animal Medicine and Surgery, University of Veterinary Medicine Hannover, 30559 Hannover, Germany; enrice.huenerfauth@tiho-hannover.de (E.H.); holger.volk@tiho-hannover.de (H.A.V.); 6Bundeswehr School of Dog Handling, 56766 Ulmen, Germany; tessakoch@bundeswehr.org (T.C.K.); wenckeschaefer@bundeswehr.org (W.S.)

**Keywords:** *Canis lupus familiaris*, neurology, neurogenetics, cerebellum, *GARNL1*, precision medicine, animal model

## Abstract

Several genetically distinct forms of cerebellar ataxia exist in Belgian shepherd dogs. We investigated a litter in which two puppies developed cerebellar ataxia. The clinical signs stabilized at around six weeks of age, but remained visible into adulthood. Combined linkage and homozygosity mapping delineated a 5.5 Mb critical interval. The comparison of whole-genome sequence data of one affected dog to 929 control genomes revealed a private homozygous ~4.8 kb deletion in the critical interval, Chr8:14,468,376_14,473,136del4761. The deletion comprises exon 35 of the *RALGAPA1* gene, XM_038544497.1:c.6080-2893_6944+1003del. It is predicted to introduce a premature stop codon into the transcript, truncating ~23% of the wild-type open reading frame of the encoded Ral GTPase-activating protein catalytic subunit α 1, XP_038400425.1:(p.Val2027Glnfs*7). Genotypes at the deletion showed the expected co-segregation with the phenotype in the family. Genotyping additional ataxic Belgian shepherd dogs revealed three additional homozygous mutant dogs from a single litter, which had been euthanized at five weeks of age due to their severe clinical phenotype. Histopathology revealed cytoplasmic accumulation of granular material within cerebellar Purkinje cells. Genotyping a cohort of almost 900 Belgian shepherd dogs showed the expected genotype–phenotype association and a carrier frequency of 5% in the population. Human patients with loss-of-function variants in *RALGAPA1* develop psychomotor disability and early-onset epilepsy. The available clinical and histopathological data, together with current knowledge about *RALGAPA1* variants and their functional impact in other species, suggest the *RALGAPA1* deletion is the likely causative defect for the observed phenotype in the affected dogs.

## 1. Introduction

Canine hereditary ataxias are a heterogeneous group of neurological diseases characterized by cerebellar or spinocerebellar dysfunction and resulting in uncoordinated movement as their hallmark [1]. Breeding practices in purebred dogs involve closed breeding populations and a certain degree of inbreeding, which increases the risk of recessively inherited diseases, including hereditary ataxias [2,3].

In Belgian shepherd dogs, congenital ataxia was first described in two puppies more than 30 years ago [4]. Spongy degeneration of the CNS led to generalized tremor and hypermetria in the two affected dogs. They had to be euthanized at two and three months of age, respectively, due to lack of improvement of clinical signs [4]. Twenty years later, a very similar phenotype was investigated in 13 Belgian shepherd puppies from five different related litters [5]. Some of these dogs were subsequently used for molecular genetic investigations of the phenotype. This led to the discovery of causative variants for spongy degeneration and cerebellar ataxia subtypes one and two (SDCA1 and SDCA2), revealing genetic heterogeneity even in clinically very comparable cases [6,7].

SDCA1 is caused by a missense variant in *KCNJ10*, encoding an inwardly rectifying potassium channel, and was reported by two independent studies [6,8] (OMIA 002089–9615). SDCA2 is caused by a SINE insertion in *ATP1B2*, coding for a subunit of a Na^+^/K^+^-ATPase [7] (OMIA 002110-9615). SDCA1 and SDCA2 exhibit strikingly similar clinical and histopathological phenotypes [7].

Another form of inherited ataxia in Belgian shepherd dogs, CNS atrophy and cerebellar ataxia (CACA), is caused by a deletion of the entire *SELENOP* gene [9] (OMIA 002367-9615). The encoded selenoprotein P is required for transport of selenium throughout the body and across the blood–brain barrier. Lastly, a frameshift variant in *SLC12A6* causes a form of spinocerebellar ataxia with later onset and slower progression than SDCA1, SDCA2, or CACA [10] (OMIA 002279-9615).

Despite recent advances in veterinary genetics, other cases of hereditary ataxia in Belgian shepherd dogs remain unsolved [11]. The current study was prompted by a report of another litter of Belgian shepherd dogs with ataxic puppies, born of parents that had been tested clear of the four known ataxia alleles in the breed. The aim of our study was to characterize the phenotype and to identify the underlying causative genetic variant.

## 2. Materials and Methods

### 2.1. Clinical Examinations

This study describes examinations in two families of Belgian shepherd dogs of the Malinois variety. At the time of the clinical and pathological investigations, it was not clear that ataxic dogs from both families shared a common genetic variant.

Family 1 originated in Austria and comprised a litter of eight puppies (3 male, 5 female). One male and one female puppy were presented for a clinical and neurological examination to the University of Veterinary Medicine, Vienna.

Family 2 originated in Germany and comprised three affected puppies (2 male and one female) that were presented for a clinical and neurological examination to the University of Veterinary Medicine Hannover. MRI examinations were performed using a 3.0 Tesla magnet (Achiever, Philips Medical Systems, Hamburg, Germany) with a standard protocol including 3D T1-weighted pre- and post gadolinium, T2-weighted transverse, dorsal and sagittal, and transverse T2-FLAIR images. Cerebrospinal fluid analysis, routine hematology and serum biochemistry were also performed.

### 2.2. Necropsy and Histology

The three affected puppies (2 male, 1 female) of family 2 were euthanized and submitted to the Department of Pathology, University of Veterinary Medicine Hannover for a full post mortem examination. Representative organ samples were taken from each dog and formalin-fixed for a minimum of 24 h, followed by paraffin embedding and cutting of 2–3 µm-thick sections using a microtome. Sections were stained with hematoxylin and eosin (H&E). In addition, selected sections of the cerebellum were stained with luxol fast blue (LFB)–cresyl violet for detection of phospholipids, Ziehl–Neelsen (ZN) for recognizing acid-fast organisms, Alcian blue to reveal the presence of acidic polysaccharides, and periodic acid–Schiff (PAS) for the detection of polysaccharides, as described previously [12].

### 2.3. Animals for Genetic Analysis

The genetic investigations were conducted on a total of 894 Belgian shepherd dog samples. We adhered to the FCI-approved (European) breed nomenclature that considers Belgian shepherd dogs a single breed with four varieties differing in coat color and type (Malinois, Tervueren, Groenendael, Laekenois). The index family (family 1) was of the Malinois variety and consisted of two affected and six unaffected littermates together with their unaffected parents. The study further included samples of 884 additional Belgian shepherd dogs from different European countries that had previously been donated to the Vetsuisse Biobank. They included 21 dogs with ataxia due to SDCA1 [6], SDCA2 [7] or CACA [9], 36 dogs with reported ataxia of unknown origin and 827 population controls without reports of similar neurological disease. The 36 dogs with ataxia of unknown origin included the 3 affected puppies from family 2. All ataxia cases in our cohort were clear of the mutant allele at the previously described *SLC12A6*:c.178_181delins-CATCTCACTCAT variant [10].

### 2.4. DNA Extraction

We extracted genomic DNA from EDTA blood samples with a Maxwell RSC whole blood DNA kit using a Maxwell RSC instrument (Promega, Madison, WI, USA). A Maxwell RSC DNA FFPE kit was used for DNA isolation from FFPE tissue samples with the same instrument.

### 2.5. Linkage Analysis and Homozygosity Mapping

Genomic DNA from all ten dogs of family 1 was genotyped on illumina_HD canine BeadChips containing 220,853 markers (Neogen, Lincoln, NE, USA). The raw SNV genotypes are available in Appendix A. For all dogs, the call rate was >95%. Markers that were missing, non-informative, on the sex chromosomes, and had Mendel errors or a minor allele-frequency <0.05 were removed using PLINK v1.9 [13]. The 99,772 markers that remained after data pruning were analyzed for parametric linkage using an autosomal recessive inheritance model with full penetrance and a disease allele frequency of 0.5 together with the Merlin software [14].

For homozygosity mapping, we used the genotype data from the two affected dogs of family 1. Markers with missing genotypes and markers on the sex chromosome were excluded. The --homozyg and --homozyg-group options in PLINK were used to search for extended regions of homozygosity >1 Mb. The output intervals were visually matched against the linked intervals in Excel spreadsheets to find overlapping regions. All positions correspond to the UU_Cfam_GSD_1.0 reference genome assembly.

### 2.6. Whole-Genome Sequencing

An Illumina TruSeq PCR-free library with ~400 bp insert size of an affected dog from family 1 was prepared. We collected 221 million 2 × 150 bp paired-end reads on a NovaSeq 6000 instrument (35.5 × coverage). Mapping and alignment were performed as described in [15]. The sequence data were deposited under study accession PRJEB16012 and sample accession SAMEA112638879 at the European Nucleotide Archive.

### 2.7. Variant Calling and Filtering

We performed variant calling using the GATK HaplotypeCaller [16] in gVCF mode as described in [15]. To filter for private variants in the affected dogs, we used genome sequences from 929 control dogs of diverse breeds (Appendix A). Sequences derived from Belgian shepherd dogs were excluded as controls for variant filtering. Predictions of functional effects of the called variants were obtained with SnpEff software [17] together with the UU_Cfam_GSD_1.0 reference genome assembly and NCBI annotation release 106.

The integrative genomics viewer [18] was used to visually inspect the critical interval for structural variants that would not have been called by the above-described automated variant calling pipeline.

### 2.8. PCR, Fragment Size Analysis and Sanger Sequencing

An allele-specific PCR assay with three primers was designed for the targeted genotyping of the Chr8:14,468,376_14,473,136del4761 variant. PCR was performed for 30 cycles using a Qiagen Multiplex PCR kit (Qiagen, Hilden, Germany) in a 10 μL reaction containing 10 ng genomic DNA, 2.5 pmol each of primers F1 and F2, and 5 pmol of primer R.

Product sizes were analyzed by automated capillary gel electrophoresis on a 5200 Fragment Analyzer (Agilent, Basel, Switzerland) together with an Agilent DNF-935 reagent kit (1–1500 bp) and according to the “DNF-935-33-DNA 1-1500bp” standard assay procedures. The wild-type allele gave rise to a 409 bp amplicon (F2-R), whereas the deletion allele yielded a product size of 339 bp (F1-R).

Another allele-specific PCR assay with three alternative primers yielding shorter amplicons was designed for DNA derived from FFPE samples. In the FFPE assay, the wild-type allele resulted in an amplicon of 130 bp (FFPE_F2-FFPE_R), while the amplicon from the deletion allele had a length of 152 bp (FFPE_F1-FFPE_R). All primer sequences are given in Appendix A.

After treatment with exonuclease I and alkaline phosphatase, we sequenced PCR amplicons on an ABI 3730 DNA analyzer (Thermo Fisher Scientific, Waltham, MA, USA). Sanger sequences were analyzed using Sequencher 5.1 software (GeneCodes, Ann Arbor, MI, USA).

## 3. Results

### 3.1. Clinical Description

Family 1 originated in Austria and comprised a litter of eight puppies (three male, five female). One male and one female puppy developed cerebellar dysfunction with an early onset of clinical signs at 4 weeks of age. During the clinical examination at 5 weeks of age, both affected puppies were less alert and active compared to the unaffected littermates, but were responsive (Appendix A). They showed a wide-based ataxic gait and exaggerated gait movements. Additionally, stumbling, staggering and intention tremor were observed. Clinical signs remained stable until the 11th week of age, and both affected dogs reached adulthood without further obvious progression of their clinical signs (Appendix A). The examined dogs did not show any systemic clinical sign other than the neurological abnormalities.

Family 2 originated in Germany. Three five-week-old puppy siblings, two male and one female, were presented with a history of chronic, slowly progressive cerebellar signs. They had an unremarkable general examination. The neurological examination revealed a wide-based stance, an ambulatory cerebellovestibular ataxia, predominantly symmetrical, with mild proprioceptive deficits. Paresis was not noted. Both menace responses were absent. Occulovestibular response was normal to reduced. A mild intention tremor of the head was noted. One puppy additionally showed erratic behavior and intermittently reduced consciousness. Therefore, the neuroanatomical localization for two puppies was suspected to be the cerebellum, whereas for the last puppy a multifocal localization including the cerebellum and the forebrain was suspected. Clinical pathology results of cerebrospinal fluid and blood were unremarkable. Magnetic resonance imaging of the brain of the puppies revealed a slight reduction in presentation of the folium area of the cerebellum, seen especially in the T2 weighted sagittal images (Figure 1). Due to the progressive nature of the neurological deficits, the owner elected euthanasia for all three puppies.

### 3.2. Necropsy and Histopathological Examinations

The three affected puppies from family 2 were necropsied. Two affected puppies (one male, one female) had good nutritional status, while one male puppy was malnourished and lacking fatty tissue in the subcutis and body cavities. Macroscopic changes did not reveal any alterations in the nervous system. Histopathologically, Purkinje cells of the cerebellum of all three puppies displayed multifocal, minimal-to-moderate accumulations of intracytoplasmic, pale basophilic material of slightly granular appearance (Figure 2a). The cytoplasmic changes stained positive with LFB (Figure 2b) and negative with all other applied stains. Other minor histopathological findings comprised a mild-to-moderate, suppurative rhinitis in all three puppies and a mild suppurative lymphadenitis in a mesenteric as well as tracheobronchial lymph node of one animal.

### 3.3. Genetic Analysis

The genetic analysis was initially started with the animals from family 1. The pedigree of this family with two ataxic puppies, a male and a female, born out of unaffected parents, suggested an autosomal recessive mode of inheritance. We obtained 220 k SNV microarray genotypes from all 10 members of the family and performed parametric linkage analysis in the family as well as autozygosity mapping in the two affected puppies. Twenty-one segments on different chromosomes with a total length of ~100 Mb showed linkage with a maximum LOD score of 1.35. When combined with the results of the autozygosity analysis, four segments with a total length of ~5.5 Mb showed linkage and shared homozygous genotypes in both available cases (Appendix A).

The genome of one affected puppy was sequenced. The automated variant calling identified 2,333,108 homozygous variants with respect to the UU_Cfam_GSD_1.0 reference genome assembly. A comparison to 929 control dog genomes from other breeds yielded 956 case-specific private variants (Appendix A). Only two of them were located within the 5.5 Mb critical interval from linkage and autozygosity mapping. Both variants were located in intergenic regions and thus not likely to have a major functional effect or to cause the investigated ataxia.

Our automated variant calling pipeline considered only small variants comprising SNVs and indels of up to ~25 nucleotides. Hence, we performed an additional visual search for structural variants in the critical interval. This search revealed a single homozygous structural variant involving protein-coding exons in the critical interval. This variant, Chr8:14,468,376_14,473,136del4761, represents a deletion removing the large exon 35 of the *RALGAPA1* gene, XM_038544497.1:c.6080-2893_6944+1003del (Figure 3). Assuming unaltered splicing of the remaining exons, the deletion is predicted to result in a frameshift and premature stop codon, XP_038400425.1:(p.Val2027Glnfs*7), thereby truncating ~23% of the wild-type *RALGAPA1* open reading frame.

The deletion was genotyped in a cohort of 894 Belgian shepherd dogs. This cohort included the index family with two cerebellar ataxia-affected dogs, 21 ataxia cases with known pathogenic variants from our earlier SDCA1, SDCA2 and CACA studies [6,7,9], and 36 other unexplained ataxia cases.

In family 1, the genotypes at the deletion co-segregated with the observed phenotype as expected for a monogenic autosomal recessive mode of inheritance. None of the previously reported SDCA1, SDCA2 and CACA cases was homozygous for the *RALGAPA1* variant. However, three of the archived unexplained ataxia cases from our biobank were also homozygous for the deletion (Figure 4). These three dogs were full siblings from family 2 that had been euthanized at 5 weeks of age due to their severe clinical phenotype at that age (described in Section 3.1 and Section 3.2).

The carrier frequency in the 827 genotyped Belgian shepherd dogs without reports of cerebellar ataxia was 5.1%. None of these dogs carried the deletion in a homozygous state (Table 1).

## 4. Discussion

In the present study, we identified a ~4.8 kb genomic deletion within the *RALGAPA1* gene in Belgian shepherd dogs with cerebellar ataxia. The *RALGAPA1* gene spans approximately 250 kb on canine chromosome 8 and gives rise to multiple transcript isoforms up to 10 kb in size. The X1 transcript isoform (XM_038544497.1) encodes a protein of 2637 amino acids with a predicted weight of 296 kDa (XP_038400425.1). The deletion identified in ataxic dogs harbors a large exon of the *RALGAPA1* gene comprising 865 nucleotides and is thus likely to disrupt gene function.

The *RALGAPA1* gene encodes Ral GTPase-activating protein catalytic subunit α 1 (RalGAPA1). RalGAPA1 or RalGAPA2 can each form a heterodimer together with the scaffolding subunit RalGAPB, thus forming the RalGAP complex [19]. RalGAP in turn regulates, together with the Ral guanine nucleotide exchange factor (RalGEF), the activity of the Ras-like (Ral) proteins RalA and RalB [20]. Those two highly related G proteins have different subcellular locations, but are both involved in a vast number of central cellular processes, including tumorigenesis [21,22]. Physiologically, RalA is implicated in cellular processes, which depend on directional membrane trafficking, such as cell migration, establishment of cell polarity, neuronal polarity, and ciliogenesis. Consequently, RalA is essential for normal brain development. *Rala^-/-^* knockout mouse embryos display exencephaly, which recently has been classified as ciliopathy, between embryonic day 10.5 and 19.5 [23].

RalGAPA1 seems to be the major catalytic subunit of the RalGAP complex in neuronal tissue of humans [24]. RalGAPA1 deficiency leads to increased RalA activity, and human patients with biallelic *RALGAPA1* variants display neuromuscular abnormalities that later develop into severe psychomotor disability and early-onset epilepsy [24] (OMIM #618797). Additionally, all human patients with pathogenic variants in *RALGAPA1* showed a moderate dysplasia of the corpus callosum on MRI [24]. Knockdown of the *RALGAPA1* homologue *tulip1* in zebrafish resulted in a comparable neurological phenotype consisting of delayed brain development and a hypomorphic head [25].

Exaggerated cerebellar signs were recognized in affected puppies at four weeks of age. At this age, puppies usually become more active and mobile. Similar initial clinical signs have also been observed in other Belgian shepherd dogs with cerebellar disease [5,6,7,9]. In contrast to most previously published cases of cerebellar dysfunction, the two affected puppies from family 1 stabilized and reached adulthood without further progression. Survival into adulthood with other inherited forms of ataxia in Belgian shepherd dogs has so far only been observed in one CACA dog [9].

The time of stabilization of cerebellar signs in the two affected dogs from family 1 corresponded to the age when puppies develop more adult-like motion patterns and cerebellar differentiation and myelination completes [26].

All three puppies of family 2 showed storage of LFB-positive, granular material within the cytoplasm of Purkinje cells of the cerebellum, indicating metabolic alterations reminiscent of an inherited lysosomal storage disease. A causal relationship of these findings with the clinically observed ataxia must be considered.

The canine *RALGAPA1*:c.6080-2893_6944+1003del variant is predicted to truncate 23% of the open reading frame of the wild-type *RALGAPA1* transcript, XP_038400425.1:(p.Val2027Glnfs*7). Early stop codon variants in human patients have been shown to result in nonsense-mediated decay of affected transcripts [23]. We assume a similar loss of function for the canine variant described herein. Furthermore, the mutant allele was not found in a homozygous state in 827 additionally genotyped Belgian shepherd dogs without neurological disease. Finally, the genotypes in the index family 1 co-segregated with the phenotype, as expected for an autosomal recessive mode of inheritance. Taken together, these findings allow us to classify the *RALGAPA1*:c.6080-2893_6944+1003del variant as pathogenic according to human diagnostic criteria [27].

## 5. Conclusions

We provide an initial clinical and pathological characterization of a new hereditary ataxia in Belgian shepherd dogs and identify *RALGAPA1*:c.6080-2893_6944+1003del as the likely causative variant for the observed phenotype. To the best of our knowledge, this is the first report of domestic animals with an *RALGAPA1*-related disease. Our results enable genetic testing, which can be used to avoid the unintentional breeding of further affected dogs.

## Figures and Tables

**Figure 1 genes-14-01520-f001:**
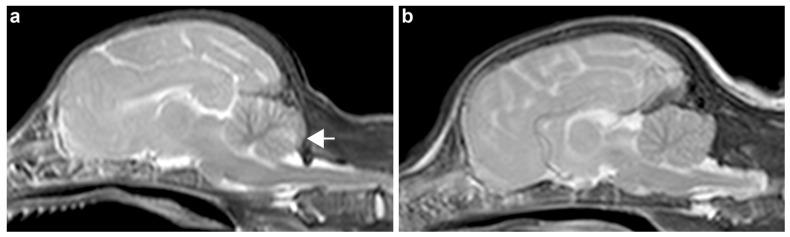
T2-weighted sagittal magnetic resonance imaging sequences of (**a**) an affected puppy from family 2 and (**b**) an age-matched control puppy without neurological deficits. (**a**) Note the different T2 weighted ill-defined hyperintensity at the level of the folium of the cerebellum in the affected puppy (arrow).

**Figure 2 genes-14-01520-f002:**
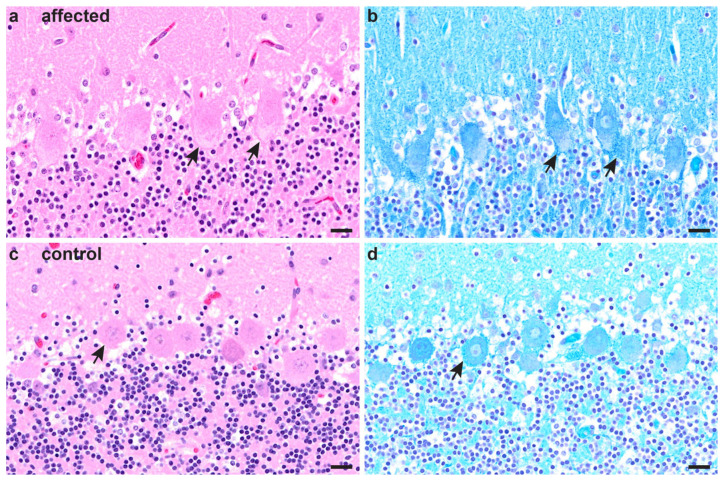
Cerebellum of an affected puppy from family 2 (**a**,**b**) and an age-matched unaffected control (**c**,**d**). (**a**) In the hematoxylin and eosin (H&E)-stained section, a lightly basophilic, slightly granular material was present in the cytoplasm of Purkinje cells (arrows). No histopathological changes were observed in the molecular or granular cell layer. (**b**) The cytoplasmic material stained blue using the luxol fast blue (LFB)–cresyl violet stain (arrows), especially around neuronal nuclei, indicating phospholipid accumulation and revealing a more distinct granular appearance of the cytoplasmic changes. (**c**) In the control, Purkinje cells show no prominent cytoplasmic changes on H&E (arrow). (**d**) The LFB stain in the control displays little cytoplasmic granular material stained blue, especially accumulating in the periphery of the perikaryon (arrow). Scale bars: 20 µm.

**Figure 3 genes-14-01520-f003:**
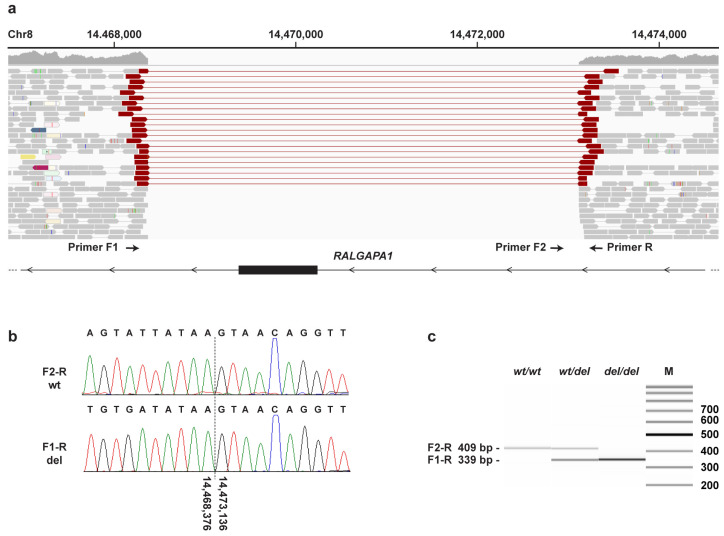
Details of the Chr8:14,468,376_14,473,136del4761 variant. (**a**) WGS short-read alignments of an affected dog indicate a homozygous deletion of 4761 bp. The deletion harbors exon 35 of *RALGAPA1*. The positions of PCR primers for an allele-specific genotyping assay are indicated (Appendix A). (**b**) Sanger sequencing of the diagnostic PCR products confirmed the deletion breakpoints. (**c**) Fragment size analysis of the PCR amplicons from genomic DNA of a healthy control (*wt/wt*), a heterozygous carrier (*wt/del*) and an affected dog (*del/del*).

**Figure 4 genes-14-01520-f004:**
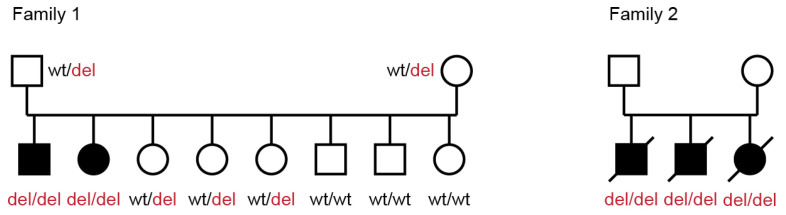
Pedigree of Belgian shepherd dogs with cerebellar ataxia. Affected dogs are indicated with filled symbols. The index family 1 with two affected dogs is shown on the left. Genotypes at the *RALGAPA1* variant are indicated for all dogs, from which a DNA sample was available. During genotyping, three additional ataxic puppies from another family (family 2) were discovered to be homozygous for the same variant. The euthanized puppies from family 2 are represented with strikethrough symbols.

**Table 1 genes-14-01520-t001:** Association of the *RALGAPA1* variant with cerebellar ataxia in 894 Belgian shepherd dogs.

Phenotype (894 Dogs)	*wt*/*wt*	*wt*/*del*	*del*/*del*
Cerebellar ataxia cases from family 1 (*n* = 2)	-	-	2
Unaffected members of family 1 (*n* = 8)	3	5	-
SDCA1, SDCA2, CACA cases (*n* = 21)	21	-	-
Unexplained archived ataxia cases (*n* = 36)	32	1	3 ^1^
Unrelated controls (*n* = 827)	785	42	-

^1^ Euthanized puppies of family 2.

## Data Availability

The accessions for the sequence data reported in this study are listed in Appendix A.

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
