# Peer review of "RALGAPA1 Deletion in Belgian Shepherd Dogs with Cerebellar Ataxia"

_genes, 2023, doi:10.3390/genes14081520_

Round 1

Reviewer 1 Report

The study was well introduced, the sampling was clearly described, and the design was adequate. The results are expressed clearly and convincingly, as are the conclusions.

The results are also clearly stated, apart from the following phrase “Four segments on different chromosomes with a total length of approx. 5.5 Mb simultaneously showed linkage with a maximum LOD score of 1.35 in the family..” (lines 221-222) that I would ask you to clarify better (particularly referring to "simultaneously" and "maximum LOD score").

Author Response

The study was well introduced, the sampling was clearly described, and the design was adequate. The results are expressed clearly and convincingly, as are the conclusions.

The results are also clearly stated, apart from the following phrase “Four segments on different chromosomes with a total length of approx. 5.5 Mb simultaneously showed linkage with a maximum LOD score of 1.35 in the family..” (lines 221-222) that I would ask you to clarify better (particularly referring to "simultaneously" and "maximum LOD score").

Response: Thank you very much for the compliments. We rephrased the confusing statement and hope that it is now easier to understand in the revised version.

Reviewer 2 Report

The author provides evidence for the RALGAPA1 deletion as a likely causative factor for the observed phenotype in the affected dogs based mostly on genetic and histology data. The article is clear, but here are a few points to be fulfilled.

Please mention the RALGAPA1 function in the introduction.

The most common sign of ataxia, regardless of the cause, is an abnormal gait in which the dog is very unsteady on his feet. Please provide pictures of the examined dogs if it is possible.

Figure 3C is good, but the original PCR gel showing the present and absent bands in wt and mutants would be preferable.

Do the dogs have other systematic symptoms?

What about the size of the cerebellum in humans and dogs?

Figure 2 is not clear, and a control image of a healthy dog should be included.

Overall, the study is useful but requires further analysis and data accuracy.

 Moderate editing of English language required

Author Response

(1)

The author provides evidence for the RALGAPA1 deletion as a likely causative factor for the observed phenotype in the affected dogs based mostly on genetic and histology data. The article is clear, but here are a few points to be fulfilled.

Response: Thank you for the positive evaluation of our manuscript.

(2)

Please mention the RALGAPA1 function in the introduction.

Response: It is our understanding that the introduction of an original research article should reflect the state of knowledge prior to the presented scientific investigation. We started this project with an unbiased hypothesis-free approach. Only during the course of the project we learned that a variant in RALGAPA1 represents a plausible candidate variant for the studied phenotype.

If we write about RALGAPA1 in the introduction, then we would incorrectly pretend that we recognized RALGAPA1 as a functional candidate gene prior to the actual analysis and that we employed a hypothesis-driven candidate gene approach. In this case, we would also have to introduce all other potential functional candidate genes for cerebellar ataxia and this would be beyond the scope of such a manuscript.

To address the comment of the reviewer, we expanded the section in the discussion, which describes the literature knowledge on RALGAPA1 function.

(3)

The most common sign of ataxia, regardless of the cause, is an abnormal gait in which the dog is very unsteady on his feet. Please provide pictures of the examined dogs if it is possible.

Response: Please refer to the supplementary videos S1 and S2. The provided videos show the dynamic gait abnormalities very clear – much better than a static picture can do this. We prefer to demonstrate the gait ataxia by the provided videos.

(4)

Figure 3C is good, but the original PCR gel showing the present and absent bands in wt and mutants would be preferable.

Response: We analyzed the PCR products by automated capillary gel electrophoresis on an Agilent 5200 Fragment Analyzer. Figure 3C represents the result of this analysis. The central image was generated by the Fragment Analyzer instrument software. This picture resembles a traditional agarose slab-gel, although it was generated by a different electrophoresis method.

Figure 3C clearly shows the presence (and absence) of specific DNA bands with precisely defined sizes according to the genotype of the investigated dogs. We do not really understand the comment of the reviewer. What else could we show? What is meant by “original PCR gel”? It would hardly make sense to show a picture of an older, less sensitive and less accurate method than what we used.

To address the comment of the reviewer, we slightly expanded the methods section and give now more details on the methodology of the fragment size analysis.

(5)

What about the size of the cerebellum in humans and dogs?

Response: Thank you for the comment. The MRI changes in the dogs were quite subtle. The size of the cerebellum did not look different in the affected puppies compared to age matched controls. In Figure 1, we highlighted with an arrow the very subtle changes seen in T2 weighted MR images, showing an ill-defined hyperintensity at the level of the folium of the cerebellum in the affected puppy. To the authors’ knowledge, changes of cerebellar size have also not been described in human patients with RALPGAP1-related disease.

(6)

Do the dogs have other systematic symptoms?

Response: The examined dogs did not show any other systemic clinical sign (not related to the neurological abnormalities). We added this to the results section.

(7)

Figure 2 is not clear, and a control image of a healthy dog should be included.

Response: We provide a revised version of Figure 2, which includes HE and LFB stainings of an age-matched unaffected control dog.

Round 2

Reviewer 2 Report

The authors made substantial revisions to this manuscript and fulfill all my comments, and I recommend article publication.

The authors made substantial revisions to this manuscript, and I recommend article publication.